



# Evaluating sensitivity of optical snow grain size retrievals to radiative transfer models, shape parameters, and inversion techniques

James W. Dillon[1], Christopher P. Donahue[2], Evan N. Schehrer[1], Kevin D. Hammonds[1]

[1]Department of Civil Engineering, Montana State University, Bozeman, MT, USA
[2]Bridger Photonics, Inc., Bozeman, MT, 59715, USA

*Correspondence to*: James W. Dillon (james.dillon013@gmail.com)

**Abstract.** The near-infrared (NIR) albedo of snow is controlled by optical snow grain size ($r_{opt}$). Therefore, characterizing the spatial and temporal variability in $r_{opt}$ at the snow surface is critical for understanding melt timing and magnitude for

water availability, and Earth's energy budget towards future climates. While numerous studies have demonstrated estimates of $r_{opt}$ via optical instruments that span scales from in situ to satellites, they leverage differing retrieval techniques, radiative transfer models, and modeled snow grain shapes. Variation in these factors cause tremendous uncertainty in $r_{opt}$ retrievals, yet a thorough evaluation has yet to be conducted. To address this knowledge gap we conducted a laboratory bidirectional reflectance study, using NIR hyperspectral imaging (NIR-HSI) to retrieve grain size metrics for a wide variety of snow

microstructures and evaluate them against micro-CT benchmarks. Towards enhanced $r_{opt}$ retrieval accuracy, we sought to determine 1) the optimal modeled snow grain shape, 2) the best-performing radiative transfer model, and 3) variability associated with retrieval techniques, spanning broadband, narrowband, multispectral, and hyperspectral approaches. Our results for optimizing grain shape parameters align with existing studies for the TARTES model, and we offer first recommendations for the SNICAR model. The retrieval technique also displayed considerable variation with the

hyperspectral residual method performing best. Multispectral and single-band techniques were comparable to their hyperspectral counterparts at times, but this was attributed to the idealized laboratory conditions and high instrument signal-to-noise ratio. Following shape-optimization, the SNICAR and TARTES models produced the best results (median absolute error of 15.6 – 17.4%, depending on technique), outperforming the AART model and the Random Mixture model. Towards a more direct comparison with albedo estimate error, we also evaluated the square root of ropt retrievals; median absolute

error values ranged from 7.9 – 26.2% depending on model and technique, with most pairings resulting in values < 15%. Our results demonstrate that the accuracy of $r_{opt}$ retrievals is highly sensitive to the choice of retrieval technique, radiative transfer model, and grain shape parameters. To minimize error, each of these factors should be carefully selected in the context of the specific measurement. As NIR-HSI instruments and other NIR detectors become increasingly affordable and their resolution improve, the findings presented here provide guidance for improved $r_{opt}$ and snow albedo mapping across ground-based,

aerial, and satellite platforms.



## 1 Introduction

Snow, the most reflective natural surface on Earth, occupies large portions of Earth's surface and plays a critical role for climate and hydrology (Dumont et al., 2021). Snow has a high (up to 90%) albedo, defined as the ratio of reflected solar
radiation at the snow surface to that of incoming solar radiation, and has a significant role in Earth's overall surface energy balance. Furthermore, snow albedo is sensitive to snow microstructure, and this sensitivity is responsible for numerous climatic feedback loops (Flanner et al., 2012). In terms of hydrology, snow albedo drives the timing and magnitude of snowmelt in mountainous regions which is imperative for water forecasting (Marks and Dozier, 1992). Thus, accurate measurements and modeled estimates of snow albedo, particularly with regards to spatiotemporal variation, are key to
understanding future climate, snowmelt rates, and water availability downstream.

The optical properties of ice are well understood (Perovich and Govoni, 1992; Picard et al., 2016; Warren and Brandt, 2008; Warren, 1982,1984), which has led to the development of numerous snow radiative transfer models used to predict the reflectance or albedo of snow based on optical conditions and physical snowpack parameters (Flanner and Zender, 2005;
Kokhanovsky and Zege, 2004; Libois et al., 2013; Malinka, 2014; Malinka et al., 2016; Stamnes et al., 1988). In the visible wavelengths, snow is highly reflective, and albedo is primarily driven by impurities near the snow surface (Skiles et al., 2012, 2018). In the near-infrared (NIR) wavelengths ice is absorptive and the primary driver of NIR albedo is the path length of ice. Historically, snow has been modeled as a collection of spheres of equivalent size (Grenfell and Warren, 1999), and thus ice path length is commonly expressed as the optical grain size (or radius), $r_{opt}$. Using this spherical assumption, the
optical grain size can then be related to the physical snow microstructure through the ice surface area per unit mass (Legagneux et al., 2002), or specific surface area (SSA). Although some models have since abandoned the spherical assumption, $r_{opt}$ remains a common means of quantifying the extent of ice absorption and SSA. Beyond NIR reflectance, optical grain size has been shown to be the primary parameter controlling broadband albedo of clean snow (Wiscombe and Warren, 1980). Therefore, characterizing the spatial and temporal variability in $r_{opt}$ at the snow surface is critical for
accurately estimating albedo from remote sensing instruments.

There is an inverse relationship between NIR albedo and optical grain size; as grain size increases, the albedo decreases due to increased absorption. This relationship is the basis from which snow reflectance measurements can be used to retrieve estimates of $r_{opt}$. A common practice is to simulate snow spectral reflectance for a wide range of $r_{opt}$ values using a radiative
transfer model and to populate a lookup table that can then be compared to measured reflectance. Over the last several decades, numerous methods have been developed to relate modeled to measured spectra. These efforts range from in situ (e.g., Donahue et al., 2021, 2022b; Gallet et al., 2009; Matzl and Schneebeli, 2006; Painter et al., 2007) to airborne platforms





(e.g., Donahue et al., 2023; Nolin and Dozier, 2000; Painter et al., 2012; Seidel et al., 2016; Skiles et al., 2023) to spaceborne sensors (e.g., Bair et al., 2020; Bohn et al., 2021; Painter et al., 2009, 2012). Although many studies have demonstrated

success at estimating $r_{opt}$, these differing methods can produce disparate retrievals. This is a salient point, as incorrect $r_{opt}$ estimates can result in substantial error in predicted snow albedo, which can dramatically influence earth system and climate models (Räisänen et al., 2017; Robledano et al., 2023). Primary sources of uncertainty or inconsistency are the data used to execute the retrieval (hereafter "retrieval technique"), the choice of radiative transfer model used, and the modeled snow grain shape used when initializing the radiative transfer model. Despite this variability, a thorough evaluation of retrieval

techniques and models has yet to be conducted.

To address these uncertainties, we conducted a laboratory reflectance study to assess $r_{opt}$ retrieval sensitivity across three factors: retrieval technique, radiative transfer model, and simulated snow grain shape. In an effort to provide future $r_{opt}$ mapping efforts with additional guidance, we sought to address the following questions:

i.    Which retrieval technique works best, and to what extent does hyperspectral data improve upon multispectral, narrowband, and broadband retrieval alternatives?
ii.   Which radiative transfer model works best?
iii.  What combination of optical snow grain shape parameters is the most effective?
iv.   How do retrieval technique, radiative transfer model, and simulated snow grain shape interplay regarding $r_{opt}$

80        retrieval accuracy?

## 2 Background

### 2.1 Retrieval techniques

The retrieval technique describes the single band, combination of bands, or spectral features used to match reflectance measurements to simulations and plays a role in grain size retrieval variability. Depending on the instrument, collected data

may be broadband, narrowband, multispectral, or hyperspectral. For broadband platforms, only a simulated average reflectance over the sensor bandwidth can be evaluated, while narrowband measurements are matched to the sensor's central wavelength. When using multispectral instruments, a normalized index, such as the Normalized Difference Grain Size Index, or NDGSI (Painter et al., 2012) is often used. Hyperspectral sensors collect continuous spectral measurements and allow for a variety of retrieval techniques, such as measuring the depth or area of normalized ice absorption features (Clark and Roush,

1984; Nolin and Dozier, 2000), or even best-match fitting the entire spectrum, known as the residual method (Donahue et al., 2022). The latter technique is useful for simultaneously retrieving grain size and liquid water content. Multispectral and hyperspectral approaches are generally considered more robust than their broadband and narrowband counterparts because they contain much more spectral information, allowing for finer discrimination of material properties and improved accuracy in detecting and characterizing specific features.





## 2.2 Radiative transfer models

In addition to differing retrieval techniques, there are several snow radiative transfer models that have been developed to simulate snow spectra, and the variability between these models also plays a key role in retrieval uncertainty. The longstanding benchmark are strict numerical codes that solve the radiative transfer equations, such as the DIScrete-Ordinate Radiative Transfer model, or DISORT (Stamnes et al., 1988). However, for many practical applications, faster and simpler approximations are often preferred. For instance, the SNow, ICe, and Aerosol Radiative – Adding-Doubling (SNICAR-AD) model (Flanner et al., 2021) is a frequently employed two-stream approximation, hence one that rapidly integrates across all viewing zenith and azimuth angles to produce albedo estimates, that has demonstrated excellent agreement with DISORT (Dang et al., 2019). Despite being an albedo model, SNICAR – AD (hereafter simply "SNICAR") is frequently compared against measured bidirectional reflectance for $r_{opt}$ retrieval at nadir viewing angles, where albedo and reflectance factor are nearly identical (Dumont et al., 2010).

The Approximate Asymptotic Radiative Transfer (AART) snow model is a bidirectional reflectance simulation based on an asymptotic approximation to the radiative transfer equation and geometric optics (Kokhanovsky and Zege, 2004). More recently, Malinka (2014) leveraged this asymptotic theory in a bidirectional reflectance model based on a random binary mixture of two immiscible materials (air and ice), in which optical characteristics change in a stochastic manner between discrete values (hereafter referred to as the "Random Mixture" model, or RM). Libois et al. (2013) combined a two-stream and asymptotic approximation scheme to create the Two-stream Analytical Radiative TransfEr in Snow (TARTES) albedo model with advanced inclusion of snow grain shape dependence. Simulated NIR snow spectra for a constant grain size, but varying radiative transfer models and shape parameters, are shown in Fig. 1 to illustrate the variability resulting from these different choices.





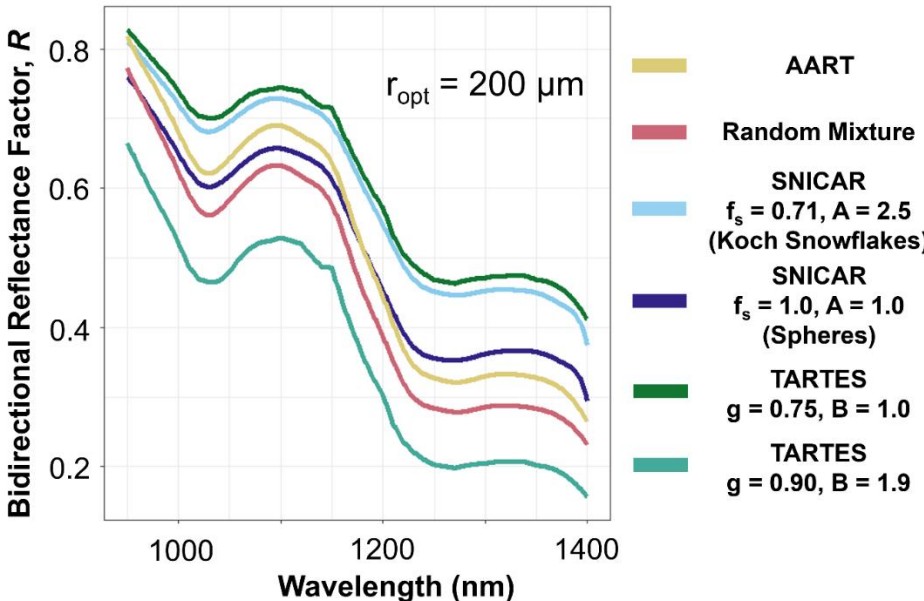

**Figure 1: Modeled snow spectra with a constant $r_{opt}$ value of 200 microns, demonstrating substantial variability between radiative transfer models and shape parameters ($f_s$, A, B, g) inputs (discussed in Sect. 2.3).**

### 2.3 Snow grain shape representation

Last, in addition to retrieval technique and radiative transfer model, another factor of relevance is the matter of modeled snow grain shape. Modeled shape, and subsequently how the single scattering grain properties are calculated, is perhaps the biggest difference between the aforementioned models, and thus the greatest cause of $r_{opt}$ retrieval uncertainty between models, and even within a given model. As discussed in Sect. 1, snow has historically been modeled as a collection of spheres of equivalent size (Grenfell and Warren, 1999), and single scattering properties determined from Mie calculations.

This was originally true of SNICAR, although the model has since been expanded to address the prevailing belief that the spherical assumption is an oversimplification. While SNICAR still calculates the single scattering albedo of snow using a spherical assumption and Mie calculations, the influence of grain shape on scattering asymmetry (specifically the asymmetry parameter, *g*) is now considered via parameterizations from He et al. (2017). The grain shape can be varied based on the combination of two parameters; the shape factor ($f_s$) and aspect ratio (A). Within the model there are four selectable

combinations of these parameters that represent idealized shapes: spheres, spheroids, hexagonal plates, and Koch snowflakes. The shape factor is defined as the ratio of the specific-projected-area-defined effective diameter of a nonspherical grain to that of a spherical grain with the same volume, representing the effect of nonsphericity (He et al., 2017). Altering the combination of $f_s$ and *A* amounts to varying the value of *g*.



The other models examined here (AART, Random Mixture, and TARTES) leverage geometric optics to calculate single scattering properties. Both AART and the Random Mixture model have fixed "shapes" (fractals and a random mixture, respectively), and thus fixed values for single scattering albedo and asymmetry parameter. The TARTES model, however, is tunable, and accounts for the influence of shape on both scattering asymmetry and absorption. Shapes in TARTES are also dependent on a two-parameter combination: the absorption enhancement parameter, $B$ (which is related to single scattering 
albedo), and the asymmetry parameter, $g$. When creating TARTES, Libois et al. (2013) called for a systemic determination of $B$ and $g$ in both the field and laboratory using independent measurements of SSA. Although the topic of modeled shape has received greater attention in recent years (e.g., He et al., 2017; Libois et al., 2013, 2014; Robledano et al., 2023), additional experiments in a controlled laboratory environment would be beneficial to the snow optics community. A summary of the models examined here and key differences between them is presented in Table 1.


**Table 1: Summary of radiative transfer model characteristics, particularly regarding modeled grain shape.**

| | Albedo/Bidirectional | Single Scattering Albedo | | Scattering Asymmetry | |
|---|---|---|---|---|---|
| | | Nonsphericity Considered | Tunable | Nonsphericity Considered | Tunable |
| SNICAR | Two-stream albedo model | No, Mie calculations | No, spheres | Yes, parameterization | Yes |
| TARTES | Two-stream albedo model | Yes, geometric optics | Yes | Yes, geometric optics | Yes |
| AART | Bidirectional | Yes, geometric optics | No, fractals | Yes, geometric optics | No, fractals |
| Random Mixture | Bidirectional | Yes, geometric optics | No, random mixture | Yes, geometric optics | No, random mixture |

In summary, variation in model and shape selection will result in substantial differences in simulated spectra (Fig. 1). These variations in modeled spectra combined with the multiple techniques used to retrieve $r_{opt}$ can lead to large error and 
uncertainty. This problem is demonstrated in Fig. 2. Three different retrieval techniques are executed across the same snow sample on a per-pixel basis. For each retrieval technique, three different models are also used to perform the retrieval, resulting in nine distinct grain size distributions that vary markedly. Despite this variability, a thorough evaluation of retrieval techniques and models has yet to be conducted, providing motivation for the evaluation presented here.





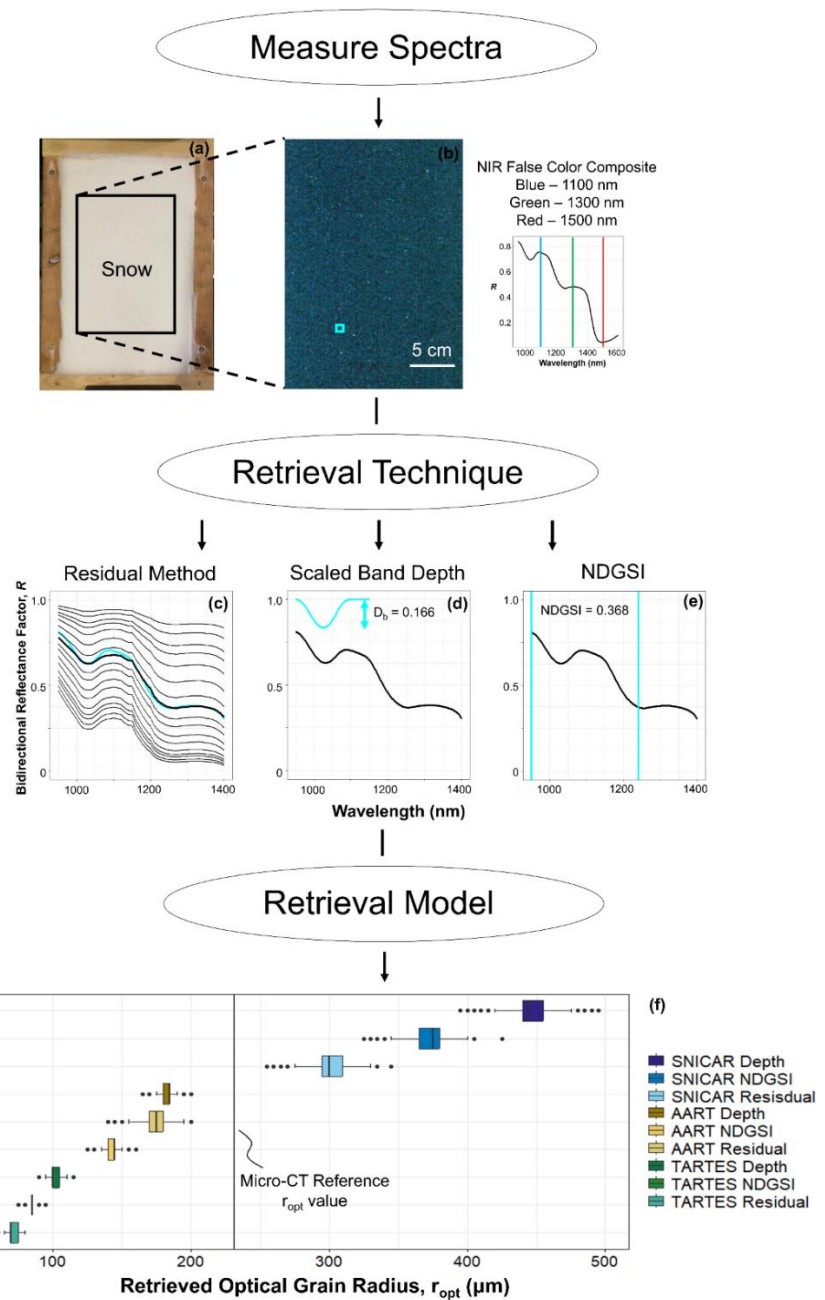

Figure 2: A demonstration of $r_{opt}$ retrieval variability across differing retrieval techniques and radiative transfer models. A visible photograph of a snow sample is shown in (a), as compared to a NIR false color composite (FCC) image in (b), produced from hyperspectral imaging. The data from the highlighted pixel (enlarged for clarity) is then evaluated using the residual (c), scaled band depth (d), and NDGSI (e) retrieval techniques, to be discussed in greater detail in Sect. 3.4. The combination of different retrieval techniques and radiative transfer models leads to dramatic differences in $r_{opt}$ retrievals (f). The black vertical reference line in (f) represents the reference micro-CT $r_{opt}$ measurement. All methods are discussed further in Sect. 3. Data shown are from Sample 18. Regarding shape, Koch snowflakes are used for SNICAR and values of 1.9 and 0.875 for $B$ and $g$, respectively, in TARTES.



## 3 Methodology

We aimed to prepare laboratory snow samples with a wide variety of well-defined grain habits and microstructures,
characterize them with microscopy and X-ray compute microtomography (micro-CT), acquire optical measurements, and
use radiative transfer modeling to perform and intercompare $r_{opt}$ retrievals. We obtained optical data using NIR hyperspectral
imaging (NIR-HSI), and determined subsequent $r_{opt}$ retrievals using numerous retrieval techniques, radiative transfer models,
and shape parameter combinations. We analyzed resulting values statistically against reference $r_{opt}$ measurements from
micro-CT. This represents one of few extensive datasets combining NIR bidirectional reflectance measurements with micro-
CT characterization of snow microstructure. Section 3.1 describes snow sample preparation and physical characterization,
Sect. 3.2 outlines the acquisition of NIR-HSI data, Sect. 3.3 covers radiative transfer modeling, and Sect. 3.4 discusses
retrieval techniques and statistical analyses. The flowchart in Fig. 3 illustrates the entirety of our retrieval comparison
process.

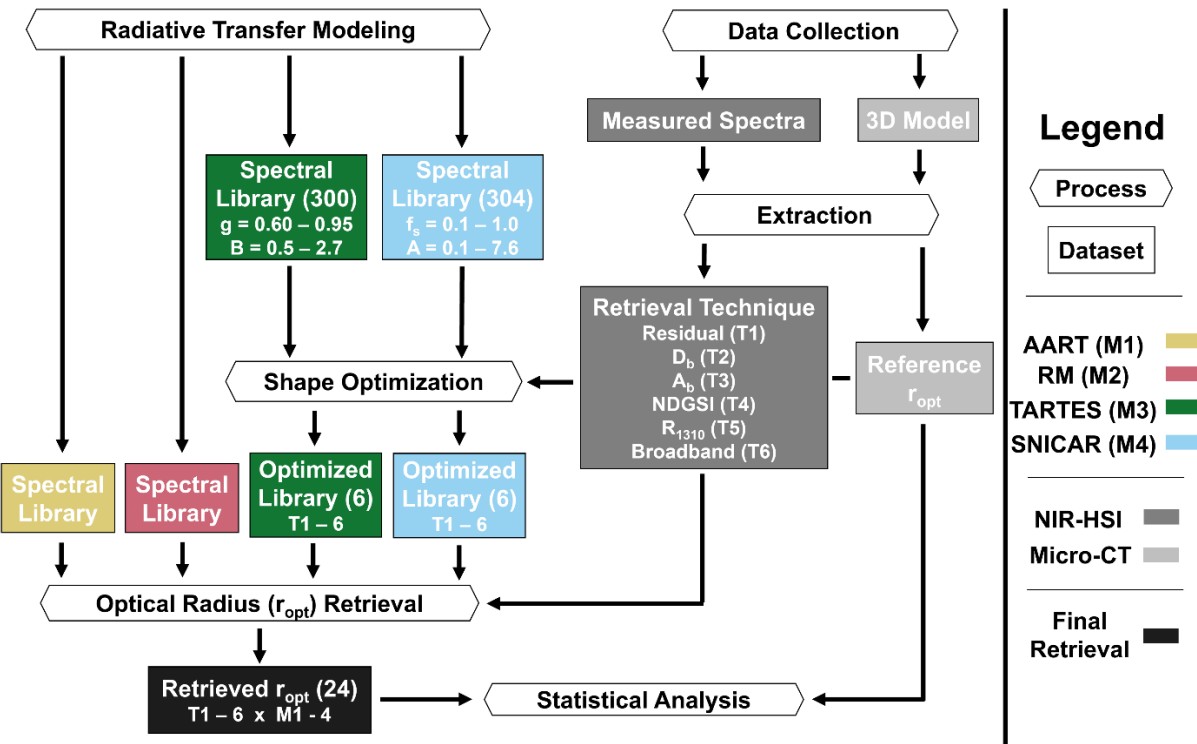

**Figure 3: Flowchart of the $r_{opt}$ retrieval and comparison process. Reflectance data from NIR-HSI were paired with four radiative transfer models (M1 – M4) to produce a variety of $r_{opt}$ retrievals. Numbers in parentheses correspond to the quantity of spectral libraries or datasets per sample. NIR-HSI data were acquired at nadir, and six different sets of data (T1 – T6) are extracted for use in retrieval. For the radiative transfer models that can consider shape, the optimal shape parameter combinations are determined for each retrieval technique. Grain size retrievals for each combination of model and retrieval technique are compared**
**on a samplewise basis to $r_{opt}$ equivalent measurements from micro-CT.**





### 3.1 Sample preparation and physical characterization

The samples used here, and thus the methods for sample preparation and physical characterization, are identical to those from Dillon et al. (2024). Sample creation and characterization are briefly summarized here, but we refer the reader to the aforementioned publication for a full description.

### 3.1.1 Sample preparation

We utilized Montana State University's Subzero Research Laboratory (SRL) for sample preparation and assessment. The snow used in these experiments was a combination of laboratory-grown crystals produced in the SRL's snowmaking apparatus and natural undisturbed snow that we collected from the surrounding area. We kept all samples in a cold room at -30° C for at least 24 hours prior to evaluation to allow them to equilibrate and ensure the snow was dry. We prepared forty-190 one snow samples from twelve batches of differing snow grains. From the bulk batches, we sieved snow grains through various mesh sizes to further promote disparate microstructures (Table 2). The exception to this was surface hoar, which we grew following the methods used by Stanton et al. (2016). Sample grain habits included precipitation particles (PP), decomposing and fragmented precipitation particles (DF), rounded grains (RG), melt forms (MF), faceted crystals (FC), depth hoar (DH), and surface hoar (SH) (Fierz et al., 2009). We prepared snow samples to be microstructurally 195 homogeneous, both laterally across the sample and vertically over sample depth of 3.8 cm.








**Table 2: Physical snow sample characteristics organized by primary grain habit and listed in order of decreasing surface area-to-volume ratio therein. Adapted from Dillon et al. (2024).**

| Sample # | Batch ID | Primary Grain Habit | Secondary Grain Habit(s) | Micro-CT SSA (kg m$^{-2}$) | Micro-CT $r_{opt}$ (μm) | Micro-CT ρ (kg m$^{-3}$) | Sieve Size (mm) Passed | Caught | Notes |
|---|---|---|---|---|---|---|---|---|---|
| 1 | A | PP | PPrm, DF | 35.85 | 91.3 | 176 | 2.38 | 1.18 | |
| 2 | A | PP | PPrm, DF | 31.60 | 103.5 | 217 | 2.38 | - | |
| 3 | A | PP | PPrm, DF | 28.69 | 114.0 | 211 | 1.18 | 0.42 | |
| 4 | B | PP | PPgp | 34.67 | 94.4 | 160 | 2.38 | 1.18 | |
| 5 | C | PP | DF | 36.10 | 90.6 | 94 | - | - | In situ fresh PP |
| 6 | C | PP | DF | 22.40 | 146.1 | 286 | 2.38 | 1.18 | |
| 7 | C | PP | DF | 22.30 | 146.7 | 280 | 0.85 | 0.42 | |
| 8 | C | PP | DF | 21.94 | 149.1 | 275 | 2.38 | - | |
| 9 | C | PP | DF | 20.05 | 163.1 | 303 | 1.18 | 0.85 | |
| 10 | D | DF | RG | 29.92 | 109.3 | 293 | 2.38 | 1.18 | |
| 11 | D | DF | RG | 28.19 | 116.1 | 323 | 0.85 | 0.42 | |
| 12 | D | DF | RG | 27.38 | 119.5 | 351 | 1.18 | 0.85 | |
| 13 | D | DF | RG | 22.65 | 144.4 | 365 | 2.38 | - | |
| 14 | E | DF | DFbk, RGwp | 17.74 | 184.4 | 374 | 0.85 | - | |
| 15 | F | DF | PP | 15.69 | 208.5 | 322 | 2.38 | - | |
| 16 | F | DF | PP | 15.04 | 217.5 | 312 | 2.38 | 1.18 | |
| 17 | F | DF | PP | 14.89 | 219.8 | 309 | 1.18 | 0.85 | |
| 18 | F | DF | PP | 14.17 | 230.9 | 382 | 0.85 | - | |
| 19 | G | FC | DH | 16.00 | 204.5 | 407 | 1.18 | 0.42 | |
| 20 | G | FC | DH | 12.34 | 265.0 | 448 | 2.38 | 1.18 | |
| 21 | G | FC | DH | 11.19 | 292.4 | 417 | 6.3 | 3.35 | |
| 22 | G | FC | DH | 10.96 | 298.5 | 472 | 6.3 | - | |
| 23 | G | FC | DH | 10.76 | 304.0 | 404 | 3.35 | 2.38 | |
| 24 | H | SH | RG | 15.83 | 206.6 | 213 | 6.3 | - | Re-sieved SH grains |
| 25 | H | SH | RG | 11.80 | 277.3 | 65 | - | - | In situ SH atop RGs |
| 26 | H | SH | RG | 8.18 | 400.0 | 94 | - | - | Smaller than S25 |
| 27 | I | RG | DF | 14.75 | 221.7 | 381 | 2.38 | 1.18 | |
| 28 | I | RG | DF | 14.26 | 229.4 | 419 | 1.18 | 0.85 | |
| 29 | I | RG | DF | 13.93 | 234.9 | 431 | 2.38 | - | |
| 30 | I | RG | DF | 13.56 | 241.4 | 489 | 0.85 | 0.42 | |
| 31 | I | RG | DF | 13.52 | 241.9 | 452 | - | - | S29 melt-refreeze |
| 32 | J | RG | DF | 15.01 | 218.0 | 394 | 0.85 | - | |
| 33 | J | RG | DF | 14.67 | 223.0 | 355 | 0.42 | - | |
| 34 | J | RG | DF | 11.62 | 281.4 | 460 | 1.18 | - | |
| 35 | K | RG | DF | 12.14 | 269.5 | 404 | 1.18 | 0.85 | |
| 36 | K | RG | DF | 11.82 | 276.8 | 428 | 0.85 | 0.42 | |
| 37 | L | MF | RG | 5.41 | 604.8 | 582 | 2.38 | 0.42 | |
| 38 | L | MF | RG | 4.02 | 813.0 | 545 | 6.3 | - | |
| 39 | L | MF | RG | 3.41 | 958.5 | 512 | 3.15 | 2.38 | |
| 40 | L | MF | RG | 3.14 | 1041.7 | 467 | - | - | Refrozen in situ |
| 41 | L | MF | RG | 2.58 | 1265.8 | 433 | 6.3 | 3.15 | |

### 3.1.2 Physical characterization

We thoroughly characterized the physical properties of each sample, as summarized in Table 2. First, we performed microscopy on representative grains from each batch prior to sieving, and classified grain habits using a crystal card and lens following Fierz et al. (2009). After sieving and sample preparation, we collected micro-CT data from each sample using a Bruker SkyScan 1173 housed in a -10° C chamber within the SRL, generally following the protocol outlined by Donahue et al. (2021). To prepare samples, we used a cylindrical holder with 3 cm diameter x 4 cm length, which allowed for a voxel size of 14.5 μm. The voxel size of 14.5 μm was the finest spatial resolution achievable with the relatively large cylindrical



micro-CT sample holder used in this study. The larger micro-CT sample holder was chosen to provide sufficient surface area for larger-grained samples (e.g., surface hoar) to be encapsulated and transported to micro-CT for measurement. We recognize that this relatively coarse resolution may lead to an underestimation of SSA and an overestimation of $r_{opt}$ for grains

with fine dendrites smaller than this size, particularly for PP primary grain habits (especially in Samples $1 - 5$).

After scanning, we performed thresholding of grey-scale images into ice and air phases by visual inspection. Reconstructions via the marching cubes method (Lorensen and Cline, 1987) allowed us to determine the volume and surface area in 3D, which we used to calculate the SSA, and thus $r_{opt}$, and density of each sample. We used these micro-CT $r_{opt}$ values as truth

for comparison to optical retrievals.

## 3.2 Hyperspectral imaging

We used a Resonon Inc. Pika NIR-640 near-infrared hyperspectral imager to map snow spectral reflectance in the NIR (www.resonon.com). Donahue et al. (2021) provide a detailed description of the instrument. Briefly, the imager's spectral resolution ranges from 2.39 to 2.50 nm, and measures 336 bands across the NIR region from 891–1711 nm. It constructs a

2D image containing the full spectrum in each pixel by collecting the image line by line, known commonly as a "push broom" scanner. We used a Resonon benchtop linear scanning stage to move the sample beneath the sensor. For more details on the benchtop apparatus, see Donahue et al. (2022).

We positioned the hyperspectral imager above the linear translating stage that held the samples. The lens of the imager is

surrounded by a set of four halogen lamps that produce direct illumination (Fig. 4a). The halogen lamps and lens of the imager are at a height of 38 and 43 cm above the snow surface, respectively. We used a large Spectralon white diffuse reflectance panel to perform calibration, resulting in a reflectance factor (R) measured for each band in every individual pixel of the image. The Spectralon panel is 30.5 x 30.5 cm, thus larger in both dimensions than our optical ROI (Fig. 4b). We built a sample holder with the same external dimensions as our snow sample holders, but specifically made to hold the Spectralon

panel, both centered on the ROI and at the same distance from the illumination source as the snow surfaces. For each snow sample scan, we also conducted a reference scan with the Spectralon panel. This allowed for pixel-by-pixel calibration of the entire optical ROI, thus accounting for any heterogeneous illumination. We made these reference measurements for each sample and each illumination condition. We acquired all optical data immediately prior to micro-CT analysis at a constant temperature of -10° C.



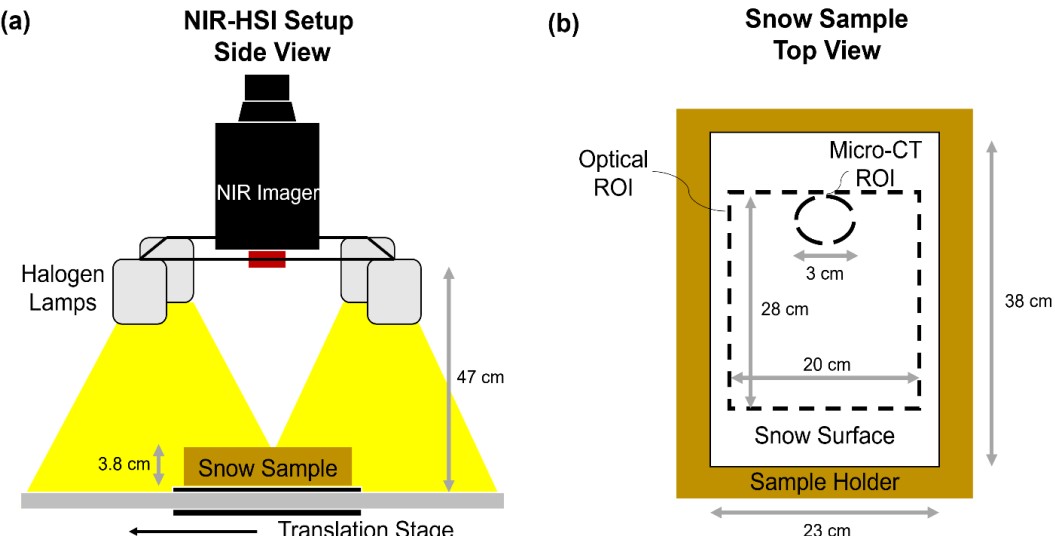


**Figure 4: Laboratory data collection schematic for hyperspectral imaging (a). Data regions-of-interest (ROIs) within the snow sample are illustrated in (b). Adapted from Dillon et al. (2024).**

Initial processing took place in Resonon's proprietary Spectronon software, and analyses thereafter performed in R. To reduce edge effects, we began by truncating each image to a central region-of-interest (ROI) that encapsulated the micro-CT

ROI (Fig. 4b). Resulting NIR-HSI ROIs contained 224,000 pixels with a spatial resolution of 0.5 mm. Reflectance images were generated from 188 of the 336 available bands, covering the range from 951 to 1403 nm. This selection was made to reduce noise at the lower end of the imager's spectral range and to exclude longer wavelengths where snow is minimally reflective. Examples of measured spectra are presented in Fig. 5. The top row depicts visible photographs of select snow samples with varying microstructures, contrasted with false color composite NIR images in the lower row. Sample spectra

from each image further demonstrate the dependence of NIR reflectance on snow microstructure.





**Figure 5: Visible photographs of several snow samples with differing microstructure (a – e) contrasted with their NIR false color composite (FCC) counterparts (f – j). Example spectra from the (enlarged) pixel in each FCC image are shown in (k), illustrating**
**the well-known relationship between grain size and reflectance in the NIR spectral range.**

## 3.3 Radiative transfer modelling

To model snow reflectance, we utilized the four commonly used snow radiative transfer models described in Sect. 2: TARTES, SNICAR, AART, and the Random Mixture model. As discussed, TARTES and SNICAR each have two tunable shape parameters which can substantially vary the modeled spectra and subsequent grain size retrievals. To further
investigate the influence of modeled snow grain shape, we produced numerous spectral libraries for both TARTES and SNICAR using modulated combinations of shape parameters. For TARTES, we evaluated absorption enhancement parameter, $B$, values from 0.8 – 2.7 at increments of 0.1, and asymmetry factor, $g$, values from 0.60 – 0.95 at increments of 0.025. These ranges spanned all reasonable values based on previous literature (e.g., Libois et al., 2013, 2014; Robledano et



al., 2023). Similarly, for SNICAR we varied the shape parameter ($f_s$) from 0.1 – 1.0 at 0.05 increments and Aspect Factor ($A$)
from 0.1 – 7.6 with steps of 0.5, again spanning all reasonable values (e.g., He et al., 2017) and nearly the full range
selectable values in the model. Thus, in total we produced 300 spectral libraries for TARTES and 304 for SNICAR, all at
nadir illumination, with each constituting a different combination of shape parameters (Fig. 3). For AART and the Random
Mixture model the modeled snow grain shape is fixed, thus we generated a single spectral library for each model with nadir
illumination and viewing angles to replicate the laboratory setup. All spectral libraries ranged from 950 – 1400 nm at 1 nm
resolution, and across $r_{opt}$ values of 30 – 1500 μm at 5 μm increments.

### 3.4 Retrieval techniques

The goal of an optical grain size retrieval is to match a measured spectrum to a modeled spectrum and obtain the quantitative
property. Therefore, to begin, all NIR-HSI data were resampled from the native spectral resolution of ~2.5 nm to 1 nm
resolution to match the modeled spectral libraries using spline interpolation. Next, we evaluated six commonly used retrieval
techniques (Fig. 6); three hyperspectral, one multispectral, one narrowband, and one broadband retrieval technique.

The first hyperspectral technique is referred to as the residual method (Donahue et al. (2022)), which leverages the entire
spectrum and minimizes the residual between the measured and modeled spectrums on a band-by-band basis (Fig. 6a). The
other two hyperspectral techniques use a spectral shape parameter related to the prominent ice absorption feature centered at
1030 nm. The scaled band depth, $D_b$ (Fig. 7b), and scaled band area, $A_b$ (Fig. 6c), approaches evaluate the continuum-
removed and normalized 1030 nm absorption feature (Clark and Roush, 1984; Nolin and Dozier, 2000). Here, the absorption
feature is defined as a range from 950 nm (fixed due to the range of the NIR-HSI instrument) to the local maxima around
1100 nm.

For a multispectral retrieval we used the NDGSI (Fig. 6d), which quantifies the relative difference between two reflectance
values in the NIR range. For our single wavelength retrieval (Fig. 6e), we selected 1310 nm, a relevant selection given its use
in the IceCube (Zuanon and A2 Photonic Sensors, 2013) and DUFISSS (Gallet et al., 2009) instruments. We also evaluated
narrowband accuracy at 1064 nm for better comparison with NIR lidar in Ackroyd (2023) and future publications (Appendix
A). Last, to emulate a broadband retrieval, we calculated the average reflectance across the entire measured spectrum (Fig.
6f).



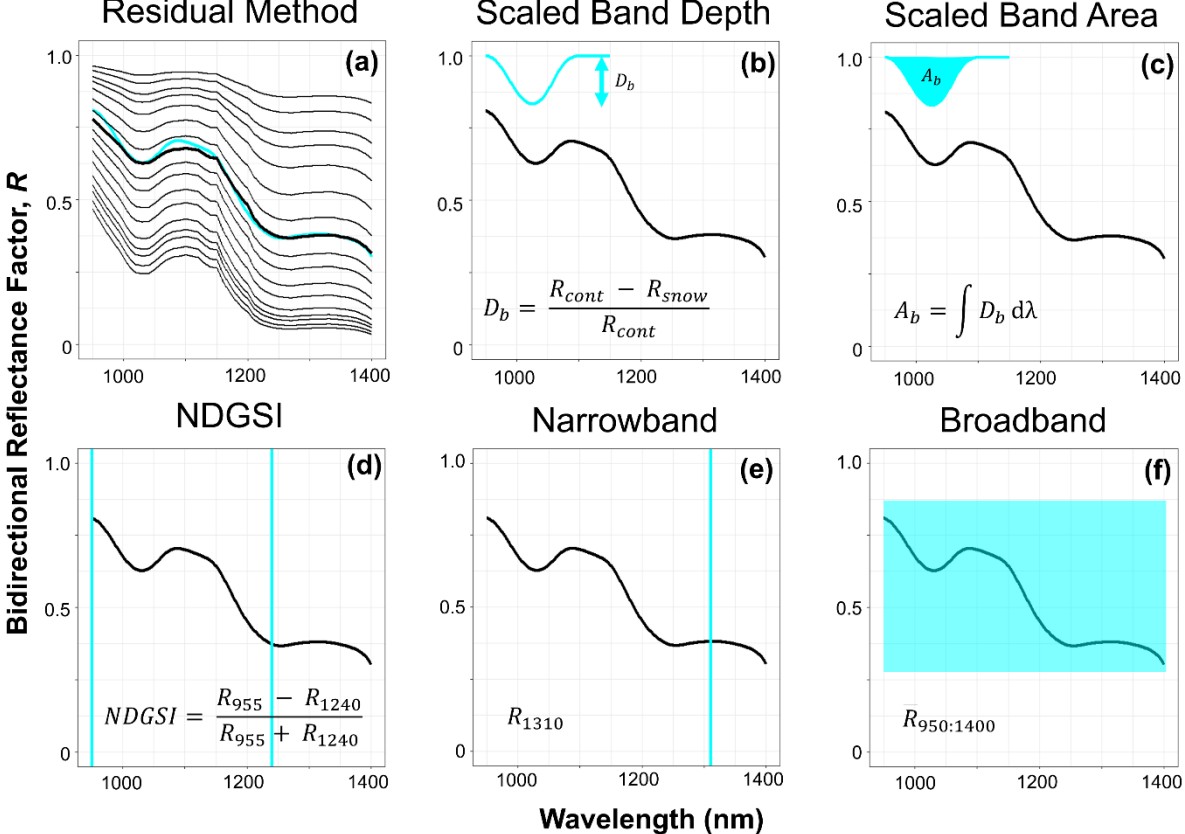

**Figure 6: Examples and definitions of the six retrieval techniques evaluated.**

For each retrieval technique, we matched the extracted data from NIR-HSI measurements to the modeled spectra with the closest corresponding value, and "retrieved" the corresponding grain size. We repeated this process for all pixels in each
sample across all spectral libraries. For the radiative transfer models that can consider shape (TARTES and SNICAR), we identified the optimal shape parameter pairing for each retrieval technique based on micro-CT measurements. We calculated samplewise medians of retrieved $r_{opt}$ and compared them to each other and across retrieval technique/model combinations, as well as to reference micro-CT values.

## 4 Results

### 4.1 Shape parameter optimization

#### 4.1.1 TARTES

Beginning with the TARTES spectral library, we calculated samplewise median values of retrieved $r_{opt}$ for each absorption enhancement/asymmetry ($B/g$) parameter combination. To visualize the influence of shape parameters, we extracted the



median absolute error (relative to micro-CT $r_{opt}$) across all samples for each technique and colored the heat map in Fig. 7 by
these error values. The optimal shape parameter combinations yielded median absolute error values of 15.5 – 17.2%, varying
slightly by retrieval technique, with hyper- and multispectral techniques generally outperforming narrow- and broadband.
However, the substantial dependence of median absolute error values on shape parameters highlights the importance of
selecting an optimal shape parameter combination.

We can see that, for a given technique, a variety of shape parameter combinations produce reasonable error (i.e., yellow
tiles). It appears that interplay between the two shape parameters is an important consideration, and thus the best selection
for one shape parameter depends on the value of the other (and, to a lesser extent, on the retrieval technique). Within the heat
maps, an interesting, yet predictable, pattern emerges in an inverse relationship between $B$ and $g$. As individual grains
become more absorptive (via an increase in $B$) accurate results are still achieved by reducing the extent to which grains
preferentially scatter forward, hence a decrease in $g$, resulting in a larger portion of the (unabsorbed) light escaping the
snowpack. While our parameter optimization for the $D_b$ retrieval technique are in good agreement with Robledano et al.
(2023), for most retrieval techniques our optimal $B/g$ combinations were closer to the idealized shapes of hexagonal plates,
cubes, cuboids, and fractals (discussed further in Sect. 5). Across all retrieval techniques, the median optimal values of $B$ and
$g$ were 1.7 ($\sigma = 0.05$) and 0.775 ($\sigma = 0.025$), respectively.

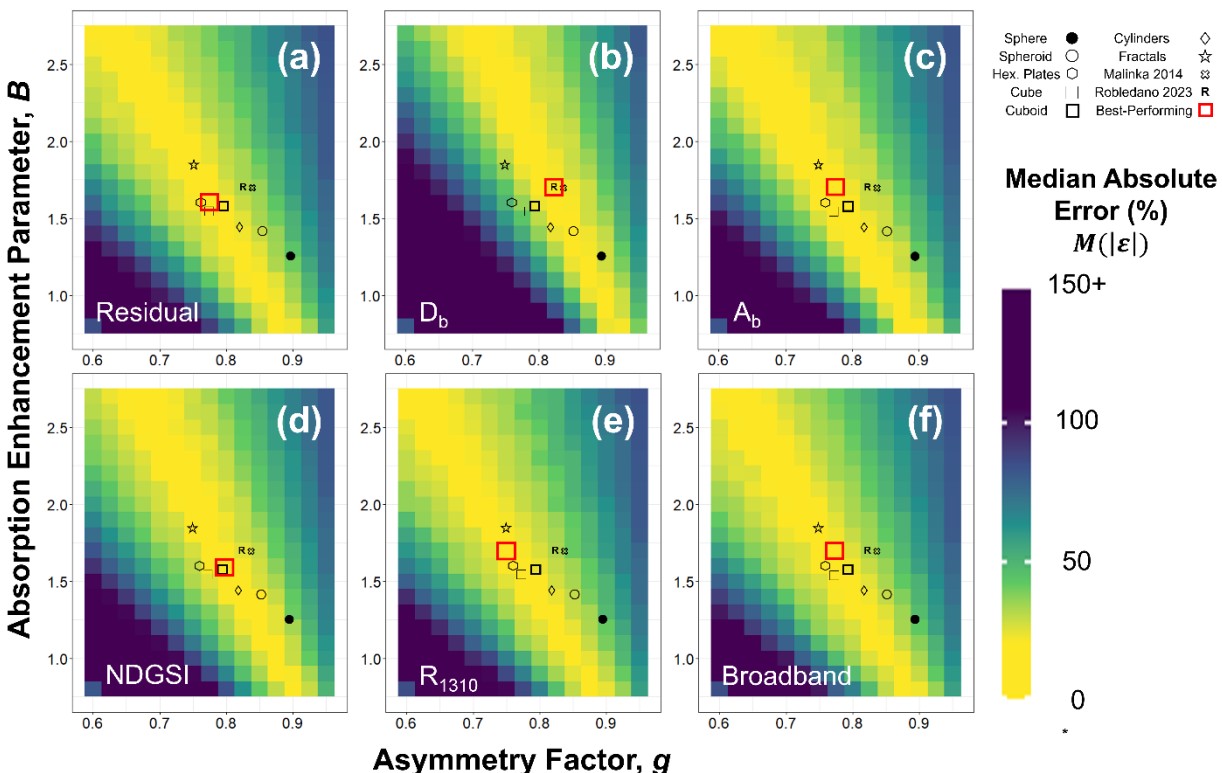






**Figure 7: Heat maps depicting median r$_{opt}$ absolute retrieval error for TARTES as a function of shape parameters, across retrieval techniques (a – f). The best-performing combination tile for each technique is boxed in red, while the optimal combination from Robledano et al. (2023) is marked with an "R", as well as other idealized shapes evaluated in their work.**

### 4.1.2 SNICAR

We performed the same heat map optimization analysis on shape parameter combinations in SNICAR (Fig. 8). The optimal shape parameter combinations yielded median absolute error values of 16.5 – 17.7%, values very comparable to TARTES. Again, significant shape dependence and patterns of optimal accuracy are apparent in the heat maps. To reiterate a key point from Sect. 2.3, unlike TARTES where the effect of shape on both absorption and asymmetry is considered, in SNICAR a spherical assumption is built into the single scattering albedo (and thus $B$). Therefore, altering the combination of shape

factor, $f_s$, and aspect ratio, $A$, is essentially akin to modulating $g$, while the value of $B$ stays fixed at that of a sphere (hence 1.25; Fig. 7). However, as we can see in Fig. 7, even for spherical values of $B$, there are corresponding values of $g$ that fall within the "stripe" of optimal accuracy in TARTES, and thus it is perhaps unsurprising that certain combinations of $f_s/A$ can yield similar retrieval accuracy in SNICAR. The optimal combination was often somewhere between the idealized shapes of spheres, spheroids, hexagonal plates and fractals (Fig. 8) as described by He et al. (2017). Across all retrieval techniques, the

median optimal values of $f_s$ and $A$ were 0.95 and 2.1, respectively, essentially amounting to an elongated spheroid.



**Figure 8: Heat maps depicting median $r_{opt}$ absolute retrieval error for SNICAR as a function of shape parameters, across retrieval techniques (a – f). The best-performing combination tile for each technique is boxed in red, while the locations of idealized shapes from He et al. (2017) are denoted as well.**

We can further observe the importance of modeled snow grain shape by comparing samplewise retrievals from SNICAR across the four pre-selected shapes using the residual method (Fig. 9). The modeled shape strongly influences both overall error and variance, with optimized shape parameters (Fig. 9e) outperforming all pre-selected shapes. Even for the optimized case, we can see that it is difficult to correctly retrieve $r_{opt}$ for different measured grain habits (particularly SH, FC, and MF) simultaneously. As expected, some shape parameter combinations fit observed grain habits better than others. Optimized

shape parameters for each model/retrieval technique are used hereafter in Sect. 4.2.



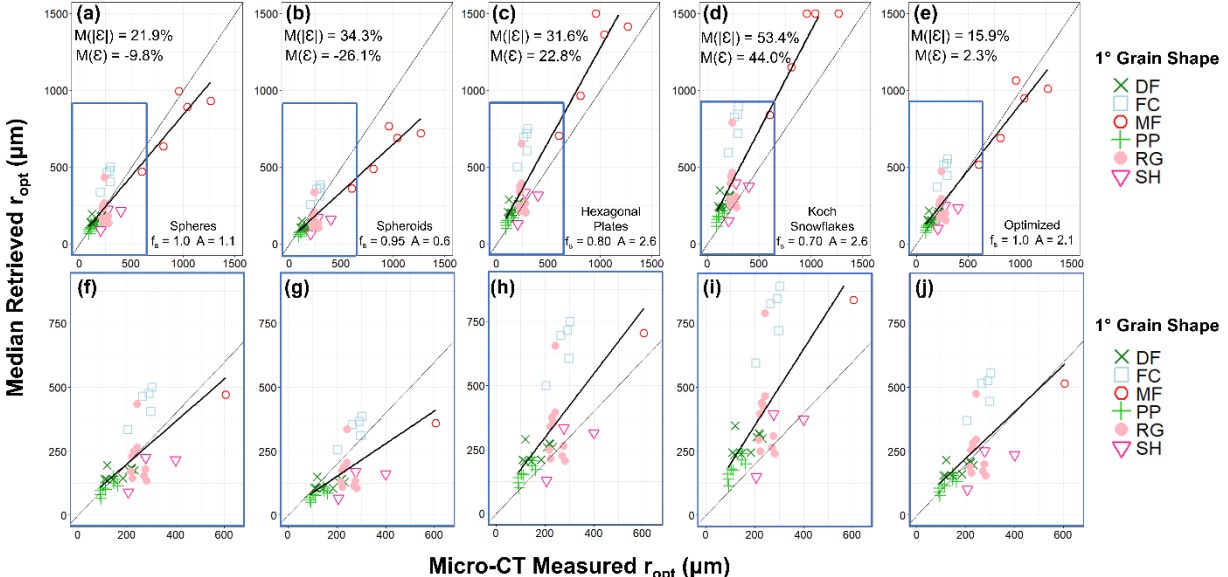

**Figure 9: Samplewise median SNICAR retrieved $r_{opt}$ values vs. Micro-CT measured $r_{opt}$ values for different preselected shapes (a – d) and optimized shape parameters (e) using residual method retrieval technique. Grey diagonal lines are a 1:1 reference while the black lines are linear best fits. Point color and style correspond to observed grain habits, following Fierz et al. (2009). The area within the blue rectangles in (a – e) is enlarged in (f – j) with resampled trendlines, given that most samples were clustered at smaller grain sizes relative to the largest MF samples. Median error and absolute error across all samples are listed for each case. For the top row, r = 0.91 in all cases, while r = 0.66 across the bottom row.**

## 4.2 Model and retrieval technique intercomparison

For a given radiative transfer model, we generally observed the most accurate results using retrieval techniques that leverage more spectral data, with reductions associated with techniques using fewer spectral data, demonstrated by the three techniques shown in Fig. 10a – 10c. This is a predictable result, although it should be noted that reductions in accuracy with fewer spectral data were often modest (e.g., TARTES and SNICAR in Fig. 10c), and in select cases even matched or outperformed their hyperspectral counterparts. Complete metrics of median absolute error for all models and retrieval techniques are provided in Table 3. Additionally, Table 3 lists median absolute error for the square root of $r_{opt}$ retrievals (lower half), considering a recent shift within the snow optics community regarding error reporting. Whereas $r_{opt}$ has a nonlinear influence on NIR absorption and albedo, the square root of $r_{opt}$ is much more linearly related to reflectance, and thus percent error in the square root of $r_{opt}$ retrievals can be directly related to uncertainty in subsequent albedo estimates.





**Figure 10: Violin and boxplots demonstrating distributions of samplewise median error across all models for the residual method**
**(a), scaled band depth (b) and $R_{1310}$ (c) retrieval techniques. In (d) probability density functions for the ratio of retrieved to**
**reference $r_{opt}$ values are shown as another means of visualizing accuracy. The right-skewing tails are largely the result of a**
**tendency to overestimate the grain size of FC samples.**



**Table 3: Median absolute error statistics (in microns) across all models and retrieval techniques for r$_{opt}$ (top) and square root r$_{opt}$ (bottom). For each model, the most accurate retrieval technique is boldened.**

| | TARTES | SNICAR | AART | RM | |
|---|---|---|---|---|---|
| Residual | 38.1 (15.9%) ± 80.0 (29.0%) | 45.1 (16.9%) ± 80.9 (30.2%) | **47.3 (20.6%) ± 84.6 (20.5%)** | **58.1 (29.8%) ± 118.0 (15.4%)** | |
| D$_b$ | 35.9 (15.6%) ± 101.4 (23.4%) | 38.0 (17.4%) ± 92.9 (24.8%) | 44.8 (21.4%) ± 123.9 (17.7%) | 66.4 (35.6%) ± 147.7 (17.2%) | $M(|\varepsilon(r_{opt})|)$ |
| A$_b$ | 35.9 (17.0%) ± 113.0 (22.3%) | 40.7 (17.7%) ± 111.9 (23.3%) | 54.4 (28.1%) ± 149.7 (15.8%) | 83.1 (44.8%) ± 177.7 (19.4%) | |
| NDGSI | **35.6 (15.5%) ± 105.6 (30.0%)** | 30.9 (17.3%) ± 106.3 (29.3%) | 64.4 (29.8%) ± 143.4 (17.5%) | 83.1 (45.5%) ± 120.0 (19.5%) | |
| R$_{1310}$ | 45.7 (17.2%) ± 87.4 (31.3%) | **44.8 (16.5%) ± 88.5 (31.9%)** | 58.1 (28.3%) ± 120.2 (15.9%) | 83.1 (39.7%) ± 112.6 (17.3%) | |
| Broadband | 40.1 (17.1%) ± 82.8 (30.4%) | 50.6 (17.7%) ± 78.9 (29.1%) | 48.1 (22.0%) ± 92.4 (19.1%) | **65.0 (29.8%) ± 116.2 (15.6%)** | |
| Residual | 1.3 (8.3%) ± 2.0 (12.4%) | **1.4 (8.3%) ± 2.0 (12.8%)** | **1.5 (10.2%) ± 1.8 (9.8%)** | **2.3 (16.2%) ± 2.2 (9.8%)** | |
| D$_b$ | 1.2 (8.1%) ± 2.1 (11.0%) | 1.4 (8.9%) ± 2.1 (11.5%) | 1.7 (11.3%) ± 2.3 (9.8%) | 2.7 (19.8%) ± 2.8 (11.1%) | $M(|\varepsilon(\sqrt{r_{opt}})|)$ |
| A$_b$ | 1.5 (8.9%) ± 2.2 (10.7%) | 1.5 (9.3%) ± 2.2 (11.2%) | 2.0 (13.4%) ± 2.8 (10.0%) | 3.3 (25.7%) ± 3.4 (12.7%) | |
| NDGSI | **1.2 (7.9%) ± 2.3 (12.8%)** | 1.2 (9.1%) ± 2.3 (12.7%) | 2.4 (16.2%) ± 2.7 (10.7%) | 3.3 (26.2%) ± 2.6 (12.6%) | |
| R$_{1310}$ | 1.4 (9.0%) ± 2.1 (13.1%) | **1.2 (8.3%) ± 2.1 (13.2%)** | 2.1 (14.2%) ± 2.2 (9.2%) | 2.9 (22.3%) ± 2.3 (11.1%) | |
| Broadband | 1.3 (8.3%) ± 2.0 (12.9%) | 1.5 (9.2%) ± 2.0 (12.4%) | 1.8 (11.7%) ± 1.8 (9.4%) | **2.3 (16.2%) ± 2.2 (10.0%)** | |

Across all retrieval techniques, a similar performance trend is apparent between models: TARTES and SNICAR produced excellent and comparable results, followed by AART, and then the Random Mixture model, a result evidenced by the violin plots in Fig. 10a – 10c, the ratio density function in Fig. 10d, and Table 3. This finding likely highlights the importance of shape optimization for a particular application and/or retrieval technique (Sect. 4.1). In other words, tuning the single scattering/inherent optical properties can be quite useful for minimizing error. An example of using the residual method and

an optimized TARTES spectral library to create pixelwise r$_{opt}$ maps for different samples is presented in Fig. 11, demonstrating the complete workflow and good agreement with micro-CT measurements.





**Figure 11: False color composite images for five different snow samples are shown in a – e. The residual method is demonstrated (f – j) on the measured spectra from the cyan pixel in each image. By repeating the process on all pixels, maps of $r_{opt}$ are created and juxtaposed with micro-CT measured values (k – o). Similarly, pixelwise grain size distributions are visualized as histograms compared to vertical micro-CT reference lines (p).**

Error metrics for the same three retrieval techniques are grouped by grain habit in Fig. 13. Much like the SNICAR scatterplot in Fig. 9, Fig. 12 demonstrates the difficulty in simultaneously producing accurate retrievals for a wide variety of snow grain habits. As with Fig. 10, we can see that TARTES and SNICAR, after shape optimization, generally perform the best, particularly with PP, DF, RG, and MF. Both AART and the Random Mixture model demonstrated a tendency to consistently underestimate grain size across most grain habits. The models generally struggled most with samples of a FC or SH primary grain habit, which is perhaps sensible, as chord lengths can vary dramatically in these crystals depending on the angle at which light interacts with the grain. More intriguing is the inconsistent sign of the error. For samples with a FC primary grain habit, all models overestimated grain size, although the Random Mixture model was quite accurate contrary to overall results. Meanwhile, SH samples were globally underestimated. While beyond the scope of this study, it would certainly be




possible to perform a similar modeled shape optimization (Sect. 4.1) towards enhancing results for a particular grain habit, if a practitioner had prior knowledge or expectation of what conditions might be encountered. However, our goal here was to optimize results across a wide range of snow microstructures because prior knowledge of grain habit is usually unknown when using remote sensing instruments.

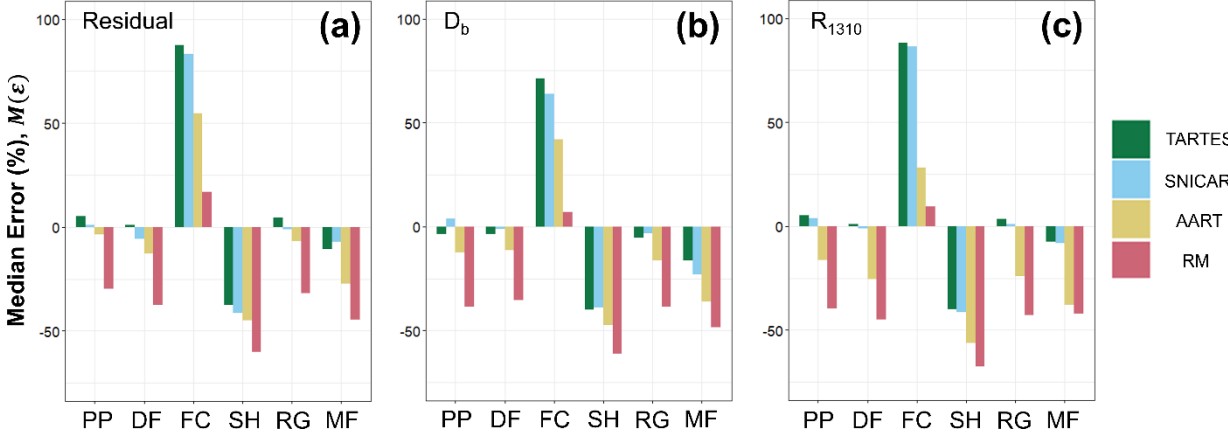


**Figure 12: Column charts grouped by grain habit depicting magnitudes of samplewise median error across all models for the residual method (a), scaled band depth (b) and R$_{1310}$ (c) retrieval techniques.**

## 5 Discussion

### 5.1 Shape optimization

Shape optimization analysis revealed that modeled snow grain shape has a substantial influence on the accuracy of grain size retrievals. For both TARTES and SNICAR, the optimal values of shape parameters were fairly constant across retrieval techniques, with similar patterns emerging (Fig. 7 and Fig. 8). Using SNICAR, optimal values of $f_s$ ranged from 0.85 – 1.00 ($M$ = 0.95). Regarding aspect ratio, $A$, values varying from unity by a factor of ~2 – 3 proved ideal (e.g., 0.6, 2.1, 2.6, 3.1). Thus, based on our results, the ideal modeled shape for SNICAR that best represents all snow grain habits is a flattened and

elongated asymmetrical spheroid. To the best of our knowledge, this is the first study to examine the optimal combination of SNICAR shape parameters and we recommend that these parameters be used in future studies.

In contrast to SNICAR, optimization of TARTES shape parameters, the absorption enhancement and asymmetry parameter ($B$ and $g$, respectively), has received considerable attention in recent years. Libois et al. (2013) suggested $1.6 \leq B \leq 1.9$,

further narrowing to the TARTES default of 1.6 in Libois et al. (2014), as they note a wide peak in their retrieved $B$ values from 1.4 – 1.8. The most recent and thorough work on the matter, conducted by Robledano et al. (2023), suggested $B = 1.7$ and $g = 0.82$, describing the optimal modeled shape of snow as, "a collection of convex particles without symmetry…". This estimate of the asymmetry parameter, $g$, deviates from the TARTES default value of 0.86 as reported by Meirold-Mautner



and Lehning (2004) and the suggested value of 0.75 from Kokhanovsky and Zege (2004); it also falls outside the range of
0.83 – 0.87 found by Libois et al. (2013). We observe asymmetry parameter values on the lower end of these observations. Our optimal $r_{opt}$ retrievals were achieved when running TARTES with $g = 0.750 – 0.825$ ($M = 0.775$) depending on the retrieval technique, thus spanning the values suggested by Kokhanovsky and Zege (2004) and Robledano et al. (2023). Regarding the absorption enhancement parameter, we observed optimal $B$ values ranging from $1.6 – 1.7$ depending on the retrieval technique, with a median value of 1.7, in agreement with Robledano et al. (2023) as well as Libois et al. (2014).


For future modeling efforts, we reiterate our median optimal shape parameters as a potential starting point: for SNICAR, $f_s = 0.95$ and $A = 2.1$; for TARTES, $B = 1.7$ and $g = 0.775$. However, there seems to be more to the story than single ideal values. We can observe in Fig. 8 and Fig. 9 that several combinations of shape parameters (for both TARTES and SNICAR) can produce similarly favorable $r_{opt}$ retrievals, and that the interplay between the two variables is most important. Though likely
difficult to enact, we recommend a similar optimization analysis for individual applications, considering instrument, retrieval technique, etc., when possible. Additionally, although certain pairings at extreme values still produced reasonable retrievals (e.g., $B = 2.7$, $g = 0.60$), we caution that these are outside the range of established values from most previous literature, and they may prove unreliable at differing illumination and viewing geometries. Furthermore, as mentioned in Sect. 4.2, it is evident that some combinations of shape parameters better represent certain grain habits than others (e.g., Fig. 9, Fig. 12).
This finding suggests that a dynamic approach, where the modeled snow grain shape is assigned based on the expected grain habits according to recent weather or seasonal conditions, would be useful, although it would require prior knowledge for effective implementation.

## 5.2 Intercomparison

Though some disagreement between optical retrievals and micro-CT measurements is to be expected, it is imperative to
understand how well optical techniques compare to true physical measurements. Such a comparison is especially important considering that broader (airborne and spaceborne) $r_{opt}$ mapping efforts are often validated by local optical retrievals (rather than micro-CT). Considering previous work, Matzl and Schneebeli (2006) found an uncertainty of 15% between SSA estimates from NIR photography and stereological measurements. In Gergely et al. (2014), grain size estimates from the Infrasnow integrating sphere demonstrated agreement within 25% relative to micro-CT based on seven of ten samples.
Gallet et al. (2009) were able to estimate SSA with error as low as 10 - 12% using their DUFISS instrument and an empirical reflectance relationship. Donahue et al. (2021) used the scaled band area retrieval technique and a hyperspectral imager to map $r_{opt}$ on a per-pixel basis in a cold laboratory. When comparing mean $r_{opt}$ retrievals to five micro-CT measurements on a semi-homogeneous sample, it was found that micro-CT measurements were 23.9% larger on average. Many of these studies used a spherical modeled grain shape, and they report $r_{opt}$ underestimations similar to those found here when using SNICAR
spheres (Fig. 9a, 9f), consistent with many papers discussing the limitations of a spherical assumption (e.g., Kokhanovsky and Zege (2004), Libois et al. (2013), Malinka et al. (2014), Robledano et al. (2023)).



Once optimized shape parameter values were applied, our results depended primarily on the radiative transfer model used, and, to a lesser extent, on the retrieval technique (Fig. 10 and Fig. 12). As discussed in Sect. 4.2, the residual method was the most accurate hyperspectral retrieval technique and often the best overall performer, a sensible result considering the
superior amount of spectral data leveraged. However, particularly when using TARTES and SNICAR, excellent results were still achieved with the multispectral, narrowband, and pseudo-broadband techniques (e.g., Fig. 10c and Table 3). This is likely due to the consistent illumination source and idealized laboratory condition; scaled absorption feature techniques were primarily introduced to limit uncertainty from low SNR and varying illumination conditions (Nolin and Dozier, 2000). However, this result is still encouraging for broadband and multispectral applications as instrument SNR and calibration
methods continue to improve.

Regarding radiative transfer models, as mentioned earlier, TARTES and SNICAR performed the best, with median absolute error ranging from 15.6 – 17.4% (or 7.9 – 9.3% for the square root of $r_{opt}$) depending on the retrieval technique, and median error of -3.5 – 5.2%. Thus, our results are either on par with or improved compared to previous literature, particularly in
relation to applications with mapping/scalable capacity as opposed to in situ techniques. The AART model followed, with median error values ranging from -29.8 – -7.1%, and then the Random Mixture model, with median error of -45.2 – -29.8%. Though we did not go so far as to hypothesize which models would have the most success, this last result is perhaps surprising, as the novel approach put forth by Malinka (2014) seems quite robust. Examining the question of modelled grain shape in terms of chord length distribution is sensible, and the model has been validated by Malinka et al. (2016), and in the
more substantial bidirectional reflectance evaluation of Dumont et al. (2021), albeit with only three snow samples spanning two grain habits in the latter. More investigation on this topic is required, as Malinka (2023) points out. For instance, the researcher demonstrates that dense packing in structures like snow, deemed only to influence light penetration depth in traditional snow radiative transfer modeling, may also result in a reduction in reflectance and albedo that has not been considered. Regardless, the efforts presented here constitute one of the most thorough comparisons between optical retrievals
and micro-CT data to date. Our success highlights the importance of considering model selection, shape optimization, and retrieval technique, as well as interactions between these factors.

**5 Conclusions**

Our research demonstrates a novel intercomparison between radiative transfer models, modeled snow grain shapes, and retrieval techniques, towards mapping snow optical grain size. In essence, we found that:
i.    Shape parameter combinations of $f_s = 0.95/A = 2.1$ and $B = 1.7/g = 0.775$ performed best for SNICAR and TARTES, respectively. However, operation-specific shape optimization would be ideal.



ii.   Regarding retrieval techniques, the hyperspectral residual method performed best. Multispectral, narrowband, and "broadband" retrieval techniques produced accuracy comparable to hyperspectral techniques when using certain models, although this result should be viewed with caution given our idealized laboratory setup.

iii.  Concerning radiative transfer models, SNICAR and TARTES (after shape-optimization) generally outperformed AART and the Random Mixture model, likely due largely to their prescribed shapes.

iv.   In general, the appropriate combination of instrument, retrieval technique, and model/shape parameters is imperative.

As NIR-HSI and other NIR detectors become more economical, and as their spatial and temporal resolution become more 510   robust, the findings presented here may provide guidance for enhanced $r_{opt}$ (and thus snow albedo) mapping. Extending the work presented here to field operations will have immediate implications for Earth surface energy balance estimates and subsequent impacts on climate, hydrological, and even avalanche forecasting.

## 6 Appendix A

Results for the narrowband $R_{1064}$ alternative retrieval technique are presented below. Shape optimization results for both 515   TARTES and SNICAR are presented in Fig. A1, while overall and grain habit-wise error metrics are presented in Fig. A2. Optimized parameters and accuracy for the $R_{1064}$ retrieval technique were comparable to all other retrieval techniques presented in the main text. However, interestingly, the TARTES and SNICAR models performed slightly worse at 1064 nm relative to all other retrieval techniques, while AART and the Random Mixture model demonstrated their best results. It is our hope that these results can eventually be used for comparison with grain size retrievals from 1064 nm lidar.


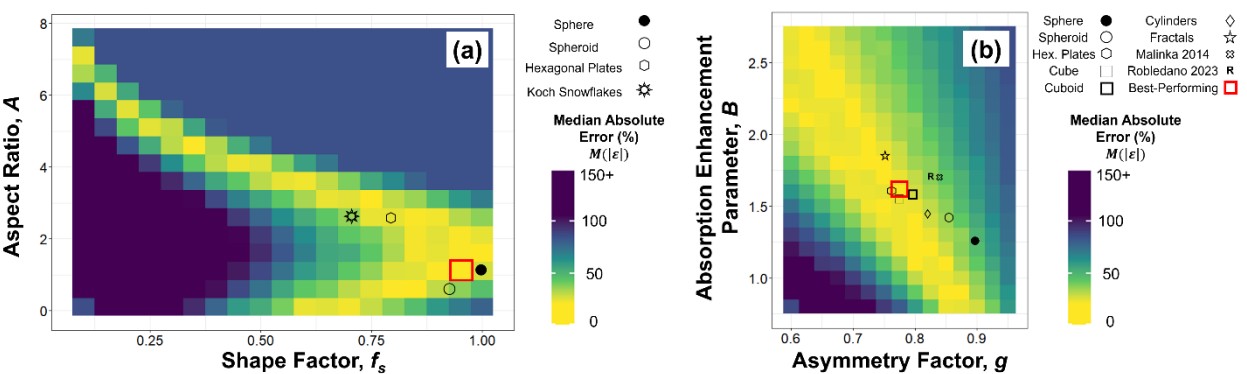

**Figure A1: Heat maps depicting median $r_{opt}$ absolute retrieval error for SNICAR (a) and TARTES (b) as a function of shape parameters for the $R_{1064}$ retrieval technique. The best-performing combination tile for each technique is boxed in red. For TARTES, the optimal combination from Robledano et al. (2023) is marked with an "R", as well as other idealized shapes 525   evaluated in their work. For SNICAR, the locations of idealized shapes from He et al. (2017) are denoted.**





**Figure A2: Violin and column graph plots depicting samplewise median error (a and b) and absolute error (c and d) for the $R_{1064}$ retrieval technique.**

**7 Data availability**

Data will be made available upon request from the lead author.



## 8 Author contribution

JD conceptualized the study, collected laboratory data, and analyzed results. CD provided guidance and advised during conceptualization and especially analysis. ES was instrumental with laboratory data collection. KH acquired funding for this research, was responsible for project administration, provided conceptual guidance, and supervised JD throughout the study. JD wrote the original draft manuscript, and all co-authors contributed during review and editing.

## 9 Competing interests

The authors declare that they have no conflict of interest.

## 10 Acknowledgements

This work was funded by the Transportation Avalanche Research Pooled Fund Program (TARP), administered through the Colorado Department of Transportation (CDOT), and by NASA Grant 80NSSC22K0694 from the Terrestrial Hydrology Program. We acknowledge the use of the Subzero Research Laboratory in the Department of Civil Engineering at Montana State University and thank Ladean McKittrick for laboratory assistance. We would like to thank Resonon, Inc. for providing us with a hyperspectral imager and technical assistance. We would also like to thank Dr. Aleksey Malinka for assistance with running the Random Mixture model. Last, we thank Joseph Shaw, Nathaniel Field and Riley Logan for technical guidance regarding optical data acquisition.

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
