# Peer review of "Evaluating sensitivity of optical snow grain size retrievals to radiative transfer models, shape parameters, and inversion techniques"

_EGUsphere, 2024_

## Author Response (AR1)

Response to reviewer 1:

Thank you very much for taking the time to read and review our manuscript. We greatly appreciate your positive feedback and valuable suggestions. Your comments will prove instrumental in improving the quality of our work. Our responses to specific comments are detailed below in red.

1. Line 24: For "ropt", the "opt" needs to be subscript.

   Good catch; we will correct this in the revised manuscript.

2. Figure 3 (Flowchart): Two missing pieces of information are the illumination condition (e.g., diffuse/direct radiation and incident/zenith angle) and snowpack conditions (e.g., snow depth and snow density) used in radiative transfer model calculations. How did the authors set these values?

   Figure 3 provides a brief overview of the methods described throughout Section 3, including radiative transfer modeling in Section 3.3, and was intended simply as an initial roadmap. However, we agree with you that these details should be stated more explicitly. We intend to add the following text to Section 3.3:

   "In order to replicate our laboratory setup, we performed simulations using direct illumination, with both illumination and viewing zenith angles equal to zero. We ran each model assuming semi-infinite snow depths, considering the depth of our sample holder was beyond the optical penetration depth for all snow microstructures and wavelengths examined here. For SNICAR, which requires a snow density input, we assigned a value of XXX kg/m3, although this was not of relevance given that density does not influence albedo or reflectance in most snow models, only penetration depth. Following the same reasoning, the other models examined here either do not require or do not allow for a density input."

3. How would the snow surface (not grain surface) roughness affect the measurements and retrieval accuracies, which is not accounted for in this study or any of the radiative transfer models?

   This is an important consideration that is receiving growing interest. It is thought that (e.g., Manninen et al., 2020) increasing snow surface roughness lowers albedo while increasing backscattering at the expense of forward scattering. Thus, in our laboratory setup, it is difficult to hypothesize whether we would see an increase or decrease (or neither) in reflectance due to dueling phenomena. We will incorporate this context, and more broadly reiterate the importance of including roughness in future measurement and modeling efforts, in our Discussion section.

4. Figure 9: for the optimized SNICAR shape parameters, what idealized shape category is used? Spheroid?

The four idealized shapes in SNICAR simply correspond to four unique combinations of the shape factor, $f_s$, and aspect ratio, A. Therefore, when a user specifies $f_s$ and A values differing from those four preset combinations, it denotes a new shape entirely (in this case the elongated spheroid discussed later), and the preset idealized shape is irrelevant. We attempted to describe this in L126 – 133 of the Background Section.

However, based on your feedback, it's obvious that this needs to be described more thoroughly to be clear to a broad audience. We will expand upon the description in Section 2.3 using the explanation above. Additionally, we will reiterate this distinction at the beginning of Section 4.1 prior to presenting shape optimization results.

5. Lines 429-431: Although the best retrieved shape in this study suggests the use of elongated asymmetrical spheroid shape, the freshly-fallen snow grain shape tends to be more like hexagonal or fractal snowflake in reality. How to reconcile this?

This is an interesting observation and topic of discussion. One would certainly think that better matching shapes should better emulate empirical bidirectional reflectance. It's possible that our asymmetrical, elongated spheroid is simply the best "middle-of-the-road" shape, optimizing performance across a wide range of observed grain habits. As we mention in L455, a dynamic approach may be prudent, where a user adjusts the simulated shape to better match the expected snow conditions, as some idealized shapes perform better with certain grain habits (such as freshly-fallen PP). However, that doesn't explain why the elongated spheroid shape produced accurate results for PP samples.

It is probably more accurate that the true "shape" of snow crystals, in an optical sense, is more complex than any idealized shape, including spheres and even fractals. Our work, and that of Robledano et al. (2023), who found the optimal shape to be "a collection of convex particles without symmetry", points to the conclusion that scattering dynamics are better captured by more abstract shape descriptions.

A third theory, presented by Malinka 2023 and discussed in L491, suggests that all current models are missing contributions from key phenomena, namely dense-packing. He suggests that this oversight may explain why certain idealized shapes, like spheres, which surely don't capture the true shape of snow crystals in most

cases, tend to produce adequate retrievals. Producing the right answer via a slightly incorrect mechanism, essentially.

In our manuscript's current form, we mention pieces of the above explanation in different sections. In the revised version, we will consolidate these observations to address this discrepancy directly (likely in Section 5.1) in order to make our findings more understandable to a broad audience.

6. It seems that the authors did not mention the uncertainties associated with different reflectance measurements, which is useful to provide.

We briefly touched on these uncertainties and the limitations of our single illumination condition when we discussed the performance of different retrieval techniques in Section 5.2, L473:

"As discussed in Sect. 4.2, the residual method was the most accurate hyperspectral retrieval technique and often the best overall performer, a sensible result considering the superior amount of spectral data leveraged. However, particularly when using TARTES and SNICAR, excellent results were still achieved with the multispectral, narrowband, and pseudo-broadband techniques (e.g., Fig. 10c and Table 3). This is likely due to the consistent illumination source and idealized laboratory condition; scaled absorption feature techniques were primarily introduced to limit uncertainty from low SNR and varying illumination conditions (Nolin and Dozier, 2000). However, this result is still encouraging for broadband and multispectral applications as instrument SNR and calibration methods continue to improve."

However, we agree that uncertainty related to illumination condition deserves its own discussion, and we will incorporate additional commentary on this. Non-nadir zenith angles and mixtures of diffuse illumination would be the norm in a field scenario, obviously. Presumably hyperspectral methods would produce more consistently accurate results in midst of changing illumination conditions, and model performance may vary as well. While we kept illumination constant to avoid introducing too many factors into our intercomparison, additional studies including different illumination conditions are greatly needed. This is all the more reason why we recommend repeating our optimization analysis for a given application, if possible.

7. Conclusion (and abstract, etc.): For the recommended optimal shape parameters for SNICAR, it also needs to mention which of the four shape categories (sphere, spheroid, hexagonal, Koch/fractal snowflake) is used.

See our response to comment 4; we will add additional explanation in Section 2.3 and reiterate in Section 4.1.

Response to Reviewer 2:

We are grateful for your review of our manuscript. We value your positive feedback and insightful suggestions, which will enhance our work. Below, we have provided detailed responses to your specific comments in red.

A related study that was omitted is *Fair et al* (2022, https://doi.org/10.5194/tc-16-3801-2022), who explored the sensitivity of retrieved snow grain size (using the band area technique) to particle shape, solar illumination and viewing geometry, and impurity concentrations. Like the present study, they found considerable sensitivity in retrieved grain size to the particle shape, albeit only through synthetic modeling studies. This study should be referenced and discussed in the context of new results presented here.

This study is very pertinent to our work; we're surprised that we hadn't come across it! We will incorporate this citation and discuss their findings as they pertain to our own in Section 5.1. Thank you for bringing this to our attention.

Section 2.2: Do all of the models used employ the same ice refractive index data in the near-IR spectrum of interest? If not, that could be one source of difference in model performance, particularly for the single-band retrieval techniques.

Good point, this is important to clarify. The ice refractive index data can be selected or input by the user for all four models. Here, we used the version from Warren and Brandt (2008) for all models/simulations. We will add a line declaring this in Section 3.3.

Figure 2: Although the y-dimension of (f) is unitless and appears to be included just for visual clarity, I suggest adding a label to the y-axis (e.g., "Model/Technique")

We agree this looks a little awkward. We will incorporate your suggestion.

Line 229: Is the micro-CT-derived $r_{opt}$ simply $3/(rho\_ice*SSA)$? Either way, please clarify this.

That is correct, and we agree that this may be vague for those less familiar with optical grain size. We will describe this relationship in L229 where we mention the conversion.

Line 276: For SNICAR (and perhaps the other models), what snowpack thickness was assumed, and what albedo of the underlying substrate was assumed? It is stated that the snow sample holder was 3.8 cm deep. For the portion of the spectrum considered in this

study, 3.8cm of snow should be optically "semi-infinite" and hence the assumed snow thickness is irrelevant (if greater than a few cm), but the issue should at least be mentioned, perhaps by informing/reminding readers of the range in penetration depth of light in snow across the wavelength range studied.

That is correct. Per this comment and similar feedback from another reviewer, we will incorporate commentary similar to the following in Section 3.3:

"In order to replicate our laboratory setup, we performed simulations using direct illumination, with both illumination and viewing zenith angles equal to zero. We ran each model assuming semi-infinite snow depths, considering the depth of our sample holder was beyond the geometric penetration depth for all snow microstructures and wavelengths examined here."

Line 277: Illumination of the snow samples in the lab arises from four lamps, and thus the illumination is not truly direct-beam nadir illumination, but something more complex. This issue seems well-accounted for in the calibration of the measurements, but since pure nadir illumination is assumed in the generation of modeled spectral libraries, there is a subtle mismatch between the illuminations applied in the measurements and modeling. It would be helpful if the authors could comment on the importance/relevance of this. Partial diffuse illumination can easily be incorporated into SNICAR and TARTES, but it is not clear how justifiable or impactful this would be. I suspect the experimental illumination is close enough to nadir that this issue is not important.

This is a good observation. Direct beam nadir illumination seemed like the most appropriate description, and we agree that this discrepancy is likely subtle enough to be negligible, though we also agree that it deserves mention. Building on the proposed addition from your previous comment:

"In order to replicate our laboratory setup, we performed simulations using direct illumination, with both illumination and viewing zenith angles equal to zero. Though our true laboratory setup is slightly more complex, with four lamps encircling the center of the snow sample, a single direct nadir beam seemed the most appropriate approximation. We ran each model assuming semi-infinite snow depths, considering the depth of our sample holder was beyond the geometric penetration depth for all snow microstructures and wavelengths examined here."

Table 3: caption: "For each model, the most accurate retrieval technique is boldened." For some models, more than one retrieval technique is boldened. Please clarify.

Multiple retrieval techniques are boldened when there was a tie for best performance, hence two techniques that produced equivalent percent error for a given model. We will clarify this.

Line 407: "Fig. 13" - There is no Figure 13.

A remnant from an older draft; good catch. Updated to read "Fig. 12".

Line 24: "opt" should be subscripted in ropt

Good catch; we will correct this in the revised manuscript.